# Significance of Tumor–Stroma Ratio (TSR) in Predicting Outcomes of Malignant Tumors

**DOI:** 10.3390/medicina59071258

**Published:** 2023-07-06

**Authors:** Jung-Soo Pyo, Nae Yu Kim, Kyueng-Whan Min, Dong-Wook Kang

**Affiliations:** 1Department of Pathology, Uijeongbu Eulji University Hospital, Eulji University School of Medicine, Uijeongbu-si 11759, Republic of Korea; jspyo@eulji.ac.kr (J.-S.P.); kyueng@eulji.ac.kr (K.-W.M.); 2Department of Internal Medicine, Uijeongbu Eulji Medical Center, Eulji University School of Medicine, Uijeongbu-si 11759, Republic of Korea; naeyu46@eulji.ac.kr; 3Department of Pathology, Chungnam National University Sejong Hospital, 20 Bodeum 7-ro, Sejong 30099, Republic of Korea; 4Department of Pathology, Chungnam National University School of Medicine, 266 Munhwa Street, Daejeon 35015, Republic of Korea

**Keywords:** tumor–stroma ratio, malignant tumor, prognosis, meta-analysis

## Abstract

*Background and Objectives:* The present study aimed to elucidate the distribution and the prognostic implications of tumor–stroma ratio (TSR) in various malignant tumors through a meta-analysis. *Materials and Methods:* This meta-analysis included 51 eligible studies with information for overall survival (OS) or disease-free survival (DFS), according to TSR. In addition, subgroup analysis was performed based on criteria for high TSR. *Results:* The estimated rate of high TSR was 0.605 (95% confidence interval (CI) 0.565–0.644) in overall malignant tumors. The rates of high TSR ranged from 0.276 to 0.865. The highest rate of high TSR was found in endometrial cancer (0.865, 95% CI 0.827–0.895). The estimated high TSR rates of colorectal, esophageal, and stomach cancers were 0.622, 0.529, and 0.448, respectively. In overall cases, patients with high TSR had better OS and DFS than those with low TSR (hazard ratio (HR) 0.631, 95% CI 0.542–0.734, and HR 0.564, 95% CI 0.0.476–0.669, respectively). Significant correlations with OS were found in the breast, cervical, colorectal, esophagus, head and neck, ovary, stomach, and urinary tract cancers. In addition, there were significant correlations of DFS in breast, cervical, colorectal, esophageal, larynx, lung, and stomach cancers. In endometrial cancers, high TSR was significantly correlated with worse OS and DFS. *Conclusions:* The rate of high TSR was different in various malignant tumors. TSR can be useful for predicting prognosis through a routine microscopic examination of malignant tumors.

## 1. Introduction

Pathological examination is performed through the interpretation of glass slides with hematoxylin and eosin (H&E) staining. In the pathological assessment of malignant tumors, the primary tumor, regional lymph node, and distant metastasis are evaluated according to the American Joint Committee on Cancer (AJCC) Cancer Staging Manual [1]. Tumor differentiation, lymphovascular invasion, perineural invasion, and resection margin involvement are evaluated in daily practice. Useful parameters for predicting the patient’s prognosis should have easy identification, high reproducibility, and less discrepancy between investigators. An epithelial tumor is composed of the tumor and surrounding stroma. Interaction between tumor cells and intra- and peritumoral stroma is important in tumor progression [2]. Evaluating these interactions can be useful for understanding tumor behavior. Stroma includes various components, such as immune cells, fibroblasts, and the extracellular matrix [3,4,5,6]. The tumor–stroma ratio (TSR), defined as the proportion of tumor area in the overall tumor, has been studied as a histologic assessment [2,7,8,9,10,11,12,13,14,15,16,17,18,19,20,21,22,23,24,25,26,27,28,29,30,31,32,33,34,35,36,37,38]. TSR is assessed by microscopic observation with H&E staining and is a method that can be sufficiently evaluated in routine pathology laboratories. However, the impact of the proportion of stroma is not clear in terms of whether it accelerates or suppresses tumor progression. Recently, the prognostic implications of TSR have been exhibited for various malignant tumors [2,8,9,10,11,12,13,14,15,16,17,18,19,20,21,22,23,24,25,26,27,28,29,30,31,32,33,34,35,36,37,38,39,40,41,42,43,44,45,46,47,48,49,50,51,52,53,54,55,56,57]. In colorectal cancers, low stroma was associated with less frequent vascular and perineural invasion and distant metastasis [45]. HIF-1α was found to be highly expressed in stroma-high tumors, with correspondingly high microvessel density in colorectal cancers. There is no conclusive information on the prognostic impacts of TSR in various malignant tumors. TSR is divided into high TSR (stroma-low) and low TSR (stroma-high) by evaluation criteria. In previous studies, the evaluation criteria of TSR affected high TSR rates [2,8,9,10,11,12,13,14,15,16,17,18,19,20,21,22,23,24,25,26,27,28,29,30,31,32,33,34,35,36,37,38,39,40,41,42,43,44,45,46,47,48,49,50,51,52,53,54,55,56,57]. If the evaluation criteria are different, the prognostic implications of TSR can differ. The evaluations of TSR have been shown to have good interobserver agreement [58] but may be more influenced by criteria. In addition, there are carcinomas that require more careful evaluation, such as lung cancer, where the amount of stroma can be inherently different between histological subtypes. It is difficult to evaluate the implications of TSR from individual studies. It is important to determine the direction of further research and analysis from a comprehensive analysis. A meta-analysis study using previous literature can be useful in obtaining comprehensive information.

We investigated high TSR rates of various malignant tumors according to malignant tumor evaluation criteria. The correlations between TSR and survival were elucidated through the subgroup analysis based on malignant tumors. The high TSR rates and prognostic impact of TSR according to evaluation criteria were analyzed.

## 2. Materials and Methods

### 2.1. Published Study Search and Selection Criteria

Relevant articles were obtained by searching the PubMed database through 15 February 2023. The following keywords were used in the search: “(tumor–stroma ratio or carcinoma–stroma ratio) AND (cancer or tumor or malignancy or neoplasm or carcinoma) AND (prognosis or prognostic or survival)”. The titles and abstracts of all searched articles were screened for inclusion and exclusion. Included articles had information on the correlation between TSR and survival in malignant tumors. However, non-original articles, such as case reports and review articles, were excluded. Articles not written in English were not included in the present study. Finally, 51 eligible articles were included in the meta-analysis (Table 1). The PRISMA checklist is shown in Appendix A. In addition, we evaluated eligible studies using the Newcastle–Ottawa Scale, and the results are presented in Table 2.

### 2.2. Data Extraction

All data were extracted from 51 eligible studies [2,8,9,10,11,12,13,14,15,16,17,18,19,20,21,22,23,24,25,26,27,28,29,30,31,32,33,34,35,36,37,38,39,40,41,42,43,44,45,46,47,48,49,50,51,52,53,54,55,56,57]. Extracted data included the author’s information, study location, number of patients analyzed, and high TSR evaluation criteria. The number and survival rates of high and low TSR were also investigated. For the quantitative aggregation of the survival results, the correlation between TSR and survival was analyzed according to the hazard ratio (HR) using one of three methods. In studies that did not take note of HRs or confidence intervals (CIs), these variables were calculated from the presented data using the HR point estimate, the log-rank statistic or its *p*-value, and the O-E statistic (the difference between the number of observed and expected events) or its variance. If those data were unavailable, HR was estimated using the total number of events, the number of patients at risk in each group, and the log-rank statistic or its *p*-value. Finally, if the only useful data were in the form of graphical representations of survival distributions, survival rates were extracted at specified times to reconstruct the HR estimate and its variance under the assumption that patients were censored at a constant rate during the time intervals [59]. The published survival curves were read independently by two authors to reduce reading variability. The HRs were then combined into an overall HR using Peto’s method [60]. Two independent authors obtained all data (Pyo J.S. and Kim N.Y.).

### 2.3. Statistical Analyses

The meta-analysis was performed using the Comprehensive Meta-Analysis software package (Biostat, Englewood, NJ, USA). The high TSR rate was investigated in various malignant tumors. TSR’s prognostic impact was evaluated, dividing survival into overall survival (OS) and disease-free survival (DFS). Heterogeneity between the studies was checked by the Q and I^2^ statistics and expressed as *p*-values. Additionally, sensitivity analysis was conducted to assess the heterogeneity of eligible studies and each study’s impact on the combined effects. In the meta-analysis, because the eligible studies used various malignant tumors and populations, a random-effects model rather than a fixed-effects model was more suitable. Begg’s funnel plot and Egger’s test were used; if significant publication bias was found, the fail-safe N and trim–fill tests were also used to confirm the degree of publication bias. The results were considered statistically significant at *p* < 0.05.

## 3. Results

### 3.1. Selection and Characteristics of the Studies

A primary search using the PubMed database found 509 relevant articles. In screening and reviewing, 409 were excluded due to inapplicable or insufficient information. Among the remaining articles, 49 reports were excluded for the following reasons: non-original articles (n = 31), non-human studies (n = 5), a language other than English (n = 11), and articles including duplicated patients (n = 2) (Figure 1).

### 3.2. Prevalence of High Tumor–Stroma Ratio

The estimated high TSR rate was 0.605 (95% CI 0.565–0.644) in overall tumors (Table 3). The highest rate of high TSR was found in endometrial cancer (0.865, 95% CI 0.827–0.895). Other female genital tract cancers, the cervical and ovary cancers, showed 0.785 (95% CI 0.713–0.842) and 0.601 (95% CI 0.417–0.761), respectively. The estimated rates of colorectal, esophageal, and stomach cancers were 0.622 (95% CI 0.556–0.683), 0.529 (95% CI 0.312–0.736), and 0.448 (95% CI 0.387–0.509), respectively. Breast cancers showed a high TSR of 50.1%. Next, subgroup analysis based on criteria for high TSR was performed because eligible studies used various criteria for high TSR. The criteria ranged from 30% to 70%. The high TSR rates for <50%, 50%, and >50% cut-off subgroups were 0.624 (95% CI 0.515–0.721), 0.609 (95% CI 0.567–0.649), and 0.399 (95% CI 0.302–0.506), respectively.

### 3.3. Correlation between High Tumor–Stroma Ratio and Survival

In overall cases, high TSR was significantly correlated with better OS and DFS compared to low TSR (HR 0.631, 95% CI 0.542–0.734, and HR 0.564, 95% CI 0.476–0.669, respectively; Table 4 and Table 5). Significant correlations with OS were found in breast, cervical, colorectal, esophageal, head and neck, ovary, stomach, and urinary tract cancers. In addition, there were significant correlations of DFS in breast, cervical, colorectal, esophageal, laryngeal, lung, and stomach cancers. However, in endometrial and pancreas cancers, high TSR was significantly correlated with a worse prognosis. In subgroup analysis based on evaluation criteria, there were significant correlations between high TSR and better OS and DFS in the subgroups with criteria <50% and 50%. In the subgroup with criteria >50%, patients with high TSR had a better OS, but not DFS, compared to patients with low TSR. 

## 4. Discussion

In the present meta-analysis, the rates of high TSR were evaluated in various malignant tumors. In addition, the correlation between TSR and survival was investigated through a meta-analysis. Previous studies used variable methods for evaluating TSR. The criterion for high TSR is usually 50% through visual inspection. Therefore, a meta-analysis is more useful for understanding the prognostic implication of TSR. The present study is the first meta-analysis, to the best of our knowledge, to elucidate the prognostic impacts of TSR according to malignant tumors and evaluation criteria.

Regardless of the origin of epithelial tumors, malignant tumors initiate through invasion into the basement membrane and progress to the stroma. This process induces changes in the characteristics of the stroma, including fibroblast proliferation and extracellular matrix deposition, through the production of cytokines and enzymes with the surrounding stroma [61,62,63,64]. Therefore, in malignant tumors, the interaction between tumor cells and stroma is important [33]. Malignant tumors have intratumoral stroma and an interface with peritumoral stroma. The definition of TSR is the proportion of tumor area in the overall tumor, including the stroma. Tumors with low TSR, which have abundant stroma, are considered active interactions between tumor cells and stroma. The tumor growth and progression are associated with the tumor microenvironment [65]. However, the detailed evaluation of the tumor environment through hematoxylin and eosin staining can be limited. TSR can be considered as a simplified analysis of the interaction between tumor cells and stroma. Therefore, the assessment of TSR may be applicable for predicting the prognosis through the routine evaluation of histology.

Recently, assessments using image analyzers have been increasingly used in research and practice. Evaluation of TSR is performed through various methods, including eyeballing and the use of a digital image analyzer [15]. The assessment of TSR can be affected by multiple factors, including the discrepancy between investigators. The evaluation criteria for high TSR are yet to be elucidated. To diminish the discrepancy caused by various factors, an image analyzer is used for evaluating TSR. The value of TSR can be different according to the evaluation foci within the tumor. The evaluation area can also affect the value of TSR, and two-tier or three-tier classification can affect the prognostic impact of TSR. Further cumulative studies for the prognostic implication of TSR gradients by evaluation criteria will be needed.

Most eligible studies investigated the evaluation criterion of 50% for high TSR. However, the previous meta-analysis showed no results for evaluation criteria [66]. With the increasing cut-off for high TSR, the rate of high TSR is lowering. The rates of high TSR were 0.624, 0.609, and 0.399 in the <50%, 50%, and >50% cut-off subgroups, respectively. In the present study, patients with high TSR had better OS and DFS than those with low TSR in the <50% and 50% cut-off subgroups. However, in the subgroup with criteria >50%, patients with high TSR had a better OS, but not DFS. In the assessment of OS in criteria >50%, colorectal cancers are only included. However, in the assessment of DFS in criteria >50%, one breast cancer study and one colorectal cancer study were included. Among these studies, there was no significant correlation between high TSR and better prognosis in a study on only breast cancer [50]. In subgroup analysis, breast cancers showed a significant correlation between high TSR and better DFS (Table 5). Although there may be a difference in the degree of HR, it can be considered that there is no significant difference in the relationship with prognosis according to the criteria.

In a pathological examination, the evaluation criteria of TNM staging differ according to malignant tumors. For example, in lung cancers, the pT stage is evaluated by tumor size and invasion depth [1]. The invasive size, rather than the overall tumor size, was significantly correlated with lung adenocarcinoma [67,68]. In addition, lung adenocarcinoma includes various histologic subtypes, such as lepidic, acinar, micropapillary, papillary, and solid adenocarcinomas. Although these subtypes have variable amounts of stroma, the specific correlation between histologic subtypes and stroma amount is not clear. Evaluating the TSR of lepidic adenocarcinoma, which has similarities with the lung’s normal parenchyma, may not be easy. Ichikawa et al. reported that lung adenocarcinoma with low TSR was significantly correlated with favorable tumor behaviors [19]. However, Xi et al. reported a significant correlation between low TSR and worse prognosis [33]. Xi’s report included adenocarcinoma and squamous cell carcinoma. However, the prognostic impact based on histologic subtypes of non-small cell lung cancers could not be elucidated in that study. In a previous study, TSR was not correlated with various clinicopathological characteristics, including histologic subtypes, pT stage, pN stage, and pTNM stage [33]. 

A previous meta-analysis was reported for the prognostic roles of TSR in gastrointestinal tract cancers [66]. There were significant correlations between high TSR and better OS in colorectal, stomach, and liver cancers. However, some discrepancies are present compared to our results. In the current meta-analysis, there was no significant correlation between TSR and OS in liver cancer. The highest and lowest rates of high TSR were found in endometrial cancer (86.5%) and stomach cancer (44.8%), respectively. This discrepancy can be caused by different characteristics of malignant tumors. Endometrial intraepithelial neoplasm, which is a precursor of endometrial cancer, has less stroma compared to the tumor area. Interestingly, for endometrial carcinoma, it was shown that high TSR was significantly correlated with worse OS and DFS. However, cervical cancers showed a significant correlation between TSR and better OS and DFS. In addition, high TSR of ovarian cancers was significantly correlated with better OS. These results suggest that the biology of tumor–stroma interactions may differ amongst cancer types.

There were some limitations in the current meta-analysis. First, high TSR rates based on histologic subtypes of each tumor could not be investigated due to insufficient information. Second, each study was not described for the evaluation area or section. In addition, it is uncertain whether the evaluation foci for TSR are hot spots or representative regions. Third, a comparison between eyeballing and image analyzers could not be performed due to insufficient information on eligible studies. Fourth, we were unable to conduct analyses by criteria subgroup for high TSR for each cancer type due to insufficient information.

## 5. Conclusions

In conclusion, our results showed that high TSR rates were different between malignant tumors. High TSR was significantly correlated with better survival rates, although some malignant tumors had no correlation or opposite correlation. Our results show that endometrial and pancreatic cancers are correlated with a poor prognosis. TSR can be useful for predicting prognosis through a routine microscopic examination of malignant tumors. Further studies for standardized histopathologic criteria will be needed in the application of TSR. 

## Figures and Tables

**Figure 1 medicina-59-01258-f001:**
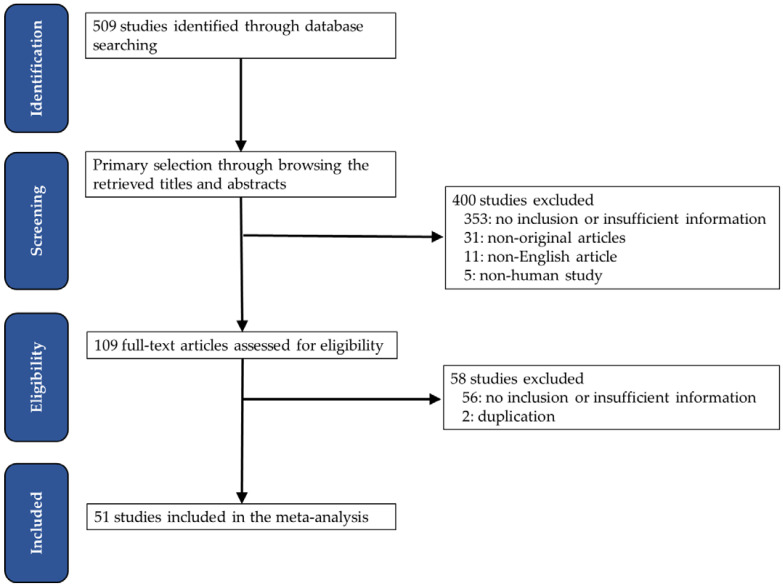
Flow chart of the searching strategy.

**Table 1 medicina-59-01258-t001:** Main characteristics of eligible studies.

Author and Publication Year	Location	Organ	Criterion for High TSR	Number of Patients	Tumor–Stroma Ratio	
					High	Low
Aboelnasr 2023 [39]	Egypt	Colorectum	50%	103	67	36
Alessandrini 2022 [40]	Italy	Larynx	50%	43	29	14
Almangush 2018 [8]	Finland/Brazil	Head and neck	50%	311	222	89
Aurello 2017 [9]	Italy	Stomach	50%	106	41	65
Chen 2015 [10]	China	Ovary	50%	838	575	263
Courrech Staal 2010 [11]	Netherlands	Esophagus	50%	93	60	33
de Kruijf 2011 [12]	Netherlands	Breast	50%	574	186	388
Dekker 2013 [13]	Netherlands	Breast	50%	403	241	162
Dourado 2020 [14]	Finland	Head and neck	50%	254	142	112
Geessink 2019 [15]	Netherlands	Colorectum	50%	129	87	42
Goyal 2021 [41]	India	Gallbladder	50%	96	56	40
Hansen 2018 [16]	Denmark	Colorectum	50%	62	33	29
He 2021 [42]	China	Esophagus	50%	270	113	157
Huang 2022 [43]	China	Head and neck	50%	151	84	67
Huijbers 2013 [17]	UK	Colorectum	50%	710	503	207
Huijbers 2018 [18]	UK	Colorectum	50%	965	642	323
Ichikawa 2018 [19]	Japan	Lung	50%	127	35	92
Inoue 2022 [44]	Japan	Colorectum	85%	200	100	100
Kairaluoma 2020 [20]	Finland	Liver	50%	47	34	13
Kang 2021 [45]	Korea	Colorectum	50%	266	185	81
Kang 2023 [46]	China	Head and neck	50%	113	56	47
Karpathiou 2019 [2]	France	Head and neck	30%	266	141	125
			50%	266	206	60
Kemi 2018 [21]	Sweden	Stomach	50%	583	241	342
Kim 2022 [47]	Korea	Stomach	40%	157	72	85
Labiche 2010 [22]	France	Ovary	50%	194	98	96
Li 2017 [23]	China	Gallbladder	50%	51	32	19
Li 2020 [48]	China	Pancreas				
Developing cohort			50%	207	120	87
Validation cohort			50%	193	112	81
Liu 2014 [24]	China	Cervix	50%	184	147	37
Lv 2015 [25]	China	Liver	50%	300	225	75
Mascitti 2020 [26]	Italy	Head and neck	50%	211	ND	ND
Öztürk 2022 [49]	Türkiye	Breast	50%	105	104	101
Panayiotou 2015 [27]	UK	Endometrium	ND	399	345	54
Peng 2018 [28]	China	Stomach	50%	494	254	240
Pongsuvareeyakul 2015 [29]	Thailand	Cervix	50%	131	93	38
Qian 2021 [50]	China	Breast	55.5%	240	93	147
Qiu 2022 [51]	China	Head and neck	50%	581	283	298
Sandberg 2018 [30]	Netherlands	Colorectum	50%	71	51	20
Scheer 2017 [31]	Netherlands	Colorectum	30%	154	118	36
			70%	154	48	106
Silva 2022 [52]	Brazil	Head and neck	50%	95	NA	NA
Smit 2020 [53]	Netherlands	Lung	50%	174	79	95
Uzun 2022 [54]	Türkiye	Gallbladder	50%	28	15	13
Vogelaar 2016 [32]	Netherlands	Colorectum	50%	97	57	40
Xi 2017 [33]	China	Lung	50%	261	223	38
Xu 2020 [34]	China	Breast	50%	260	146	114
Xu 2023 [55]	China	Urinary tract	50%	1015	622	393
Yan 2022 [56]	China	Breast	33.5%	240	153	87
Zengin 2019 [35]	Türkiye	Colorectum	50%	88	52	36
Zhang 2014 [36]	China	Head and neck	50%	93	51	42
Zhang 2015 [37]	China	Lung	50%	404	302	102
Zheng 2023 [57]	China	Urinary bladder	45.7%	133	94	39
Zong 2020 [38]	China	Cervix	50%	384	317	67

TSR, tumor–stroma ratio; UK, United Kingdom; ND, no data.

**Table 2 medicina-59-01258-t002:** Results of quality assessment using the Newcastle–Ottawa Scale for eligible studies.

Author and Publication Year	Is the Case Definition Adequate	Representativeness of the Cases	Selection of Controls	Definition of Controls	Comparability of Cases and Controls on the Basis of the Design or Analysis	Ascertainment of Exposure	Same Method of Ascertainment for Cases and Controls	Non-Response Rate	Quality Score
Aboelnasr 2023 [39]	1	1	-	1	2	1	1	1	8
Alessandrini 2022 [40]	1	1	-	1	2	1	1	1	8
Almangush 2018 [8]	1	1	-	1	2	1	1	1	8
Aurello 2017 [9]	1	1	-	1	2	1	1	1	8
Chen 2015 [10]	1	1	-	1	2	1	1	1	8
Courrech Staal 2010 [11]	1	1	-	1	2	1	1	1	8
de Kruijf 2011 [12]	1	1	-	1	2	1	1	1	8
Dekker 2013 [13]	1	1	-	1	2	1	1	1	8
Dourado 2020 [14]	1	1	-	1	2	1	1	1	8
Geessink 2019 [15]	1	1	-	1	2	1	1	1	8
Goyal 2021 [41]	1	1	-	1	2	1	1	1	8
Hansen 2018 [16]	1	1	-	1	2	1	1	1	8
He 2021 [42]	1	1	-	1	2	1	1	1	8
Huang 2022 [43]	1	1	-	1	2	1	1	1	8
Huijbers 2013 [17]	1	1	-	1	2	1	1	1	8
Huijbers 2018 [18]	1	1	-	1	2	1	1	1	8
Ichikawa 2018 [19]	1	1	-	1	2	1	1	1	8
Inoue 2022 [44]	1	1	-	1	2	1	1	1	8
Kairaluoma 2020 [20]	1	1	-	1	2	1	1	1	8
Kang 2021 [45]	1	1	-	1	2	1	1	1	8
Kang 2023 [46]	1	1	-	1	2	1	1	1	8
Karpathiou 2019 [2]	1	1	-	1	2	1	1	1	8
Kemi 2018 [21]	1	1	-	1	2	1	1	1	8
Kim 2022 [47]	1	1	-	1	2	1	1	1	8
Labiche 2010 [22]	1	1	-	1	2	1	1	1	8
Li 2017 [23]	1	1	-	1	2	1	1	1	8
Li 2020 [48]	1	1	-	1	2	1	1	1	8
Liu 2014 [24]	1	1	-	1	2	1	1	1	8
Lv 2015 [25]	1	1	-	1	2	1	1	1	8
Mascitti 2020 [26]	1	1	-	1	2	1	1	1	8
Öztürk 2022 [49]	1	1	-	1	2	1	1	1	8
Panayiotou 2015 [27]	1	1	-	1	2	1	1	1	8
Peng 2018 [28]	1	1	-	1	2	1	1	1	8
Pongsuvareeyakul 2015 [29]	1	1	-	1	2	1	1	1	8
Qian 2021 [50]	1	1	-	1	2	1	1	1	8
Qiu 2022 [51]	1	1	-	1	2	1	1	1	8
Sandberg 2018 [30]	1	1	-	1	2	1	1	1	8
Scheer 2017 [31]	1	1	-	1	2	1	1	1	8
Silva 2022 [52]	1	1	-	1	2	1	1	1	8
Smit 2020 [53]	1	1	-	1	2	1	1	1	8
Uzun 2022 [54]	1	1	-	1	2	1	1	1	8
Vogelaar 2016 [32]	1	1	-	1	2	1	1	1	8
Xi 2017 [33]	1	1	-	1	2	1	1	1	8
Xu 2020 [34]	1	1	-	1	2	1	1	1	8
Xu 2023 [55]	1	1	-	1	2	1	1	1	8
Yan 2022 [56]	1	1	-	1	2	1	1	1	8
Zengin 2019 [35]	1	1	-	1	2	1	1	1	8
Zhang 2014 [36]	1	1	-	1	2	1	1	1	8
Zhang 2015 [37]	1	1	-	1	2	1	1	1	8
Zheng 2023 [57]	1	1	-	1	2	1	1	1	8
Zong 2020 [38]	1	1	-	1	2	1	1	1	8

**Table 3 medicina-59-01258-t003:** The estimated rates of high tumor–stroma ratio in various malignant tumors.

	Number of Subsets	Fixed Effect (95% CI)	Heterogeneity Test (*p*-Value)	Random Effect (95% CI)	Egger’s Test (*p*-Value)
Overall	52	0.577 (0.588, 0.605)	<0.001	0.605 (0.565, 0.644)	0.638
Breast	6	0.483 (0.460, 0.506)	<0.001	0.501 (0.391, 0.612)	0.331
Cervix	3	0.794 (0.762, 0.823)	0.019	0.785 (0.713, 0.842)	0.375
Colorectum	12	0.647 (0.630, 0.665)	<0.001	0.622 (0.556, 0.683)	0.300
Endometrium	1	0.865 (0.827, 0.895)	1.000	0.865 (0.827, 0.895)	-
Esophagus	2	0.475 (0.423, 0.527)	<0.001	0.529 (0.312, 0.736)	-
Gallbladder	3	0.588 (0.514, 0.659)	0.722	0.588 (0.514, 0.659)	0.820
Head and neck	8	0.578 (0.556, 0.600)	<0.001	0.594 (0.513, 0.671)	0.393
Larynx	1	0.674 (0.523, 0.797)	1.000	0.674 (0.523, 0.797)	-
Liver	2	0.746 (0.698, 0.789)	0.697	0.746 (0.698, 0.789)	-
Lung	4	0.648 (0.613, 0.680)	<0.001	0.606 (0.343, 0.819)	0.544
Ovary	2	0.650 (0.620, 0.679)	<0.001	0.601 (0.417, 0.761)	-
Pancreas	2	0.580 (0.531, 0.627)	0.990	0.580 (0.531, 0.627)	-
Stomach	4	0.454 (0.427, 0.481)	0.005	0.448 (0.387, 0.509)	0.758
Urinary tract	2	0.623 (0.595, 0.651)	0.036	0.653 (0.556, 0.738)	-
Criteria					
<50%	5	0.603 (0.570, 0.634)	<0.001	0.624 (0.515, 0.721)	0.285
50%	43	0.600 (0.591, 0.609)	<0.001	0.609 (0.567, 0.649)	0.630
>50%	3	0.408 (0.368, 0.448)	0.001	0.399 (0.302, 0.506)	0.605

CI, confidence interval.

**Table 4 medicina-59-01258-t004:** The correlation between high tumor–stroma ratio and overall survival in various malignant tumors.

	Number of Subsets	Fixed Effect (95% CI)	Heterogeneity Test (*p*-Value)	Random Effect (95% CI)	Egger’s Test (*p*-Value)
Overall	40	0.657 (0.616, 0.701)	<0.001	0.631 (0.542, 0.734)	0.363
Breast	2	0.645 (0.487, 0.856)	0.284	0.630 (0.443, 0.896)	-
Cervix	3	0.377 (0.258, 0.551)	0.384	0.377 (0.258, 0.551)	0.785
Colorectum	10	0.643 (0.553, 0.747)	<0.001	0.588 (0.429, 0.804)	0.308
Endometrium	1	2.510 (1.223, 5.152)	1.000	2.510 (1.223, 5.152)	-
Esophagus	2	0.406 (0.294, 0.559)	0.854	0.406 (0.294, 0.559)	-
Gallbladder	3	0.574 (0.346, 0.954)	0.132	0.568 (0.276, 1.169)	0.152
Head and neck	4	0.610 (0.496, 0.750)	0.108	0.563 (0.400, 0.792)	0.033
Liver	2	0.503 (0.316, 0.802)	0.140	0.538 (0.262, 1.105)	-
Lung	4	0.719 (0.574, 0.900)	0.001	0.843 (0.482, 1.474)	0.246
Ovary	2	0.834 (0.711, 0.978)	0.552	0.834 (0.711, 0.978)	-
Pancreas	2	1.957 (1.443, 2.654)	0.779	1.957 (1.443, 2.654)	-
Stomach	3	0.498 (0.421, 0.589)	0.063	0.456 (0.324, 0.641)	0.214
Urinary tract	2	0.599 (0.491, 0.730)	0.048	0.636 (0.417, 0.971)	-
Criteria					
>50%	3	0.748 (0.605, 0.924)	0.603	0.748 (0.605, 0.924)	0.388
50%	34	0.639 (0.596, 0.684)	<0.001	0.593 (0.501, 0.702)	0.206
>50%	2	0.728 (0.483, 1.095)	0.378	0.728 (0.483, 1.095)	-

CI, confidence interval.

**Table 5 medicina-59-01258-t005:** The correlation between high tumor–stroma ratio and disease-free survival in various malignant tumors.

	Number of Subsets	Fixed Effect (95% CI)	Heterogeneity Test (*p*-Value)	Random Effect (95% CI)	Egger’s Test (*p*-Value)
Overall	29	0.571 (0.522, 0.623)	<0.001	0.564 (0.476, 0.669)	0.551
Breast	4	0.517 (0.423, 0.632)	0.313	0.517 (0.415, 0.645)	0.848
Cervix	3	0.447 (0.307, 0.650)	0.519	0.447 (0.307, 0.650)	0.351
Colorectum	6	0.609 (0.518, 0.716)	0.200	0.609 (0.490, 0.759)	0.826
Endometrium	1	2.180 (1.146, 4.146)	1.000	2.180 (1.146, 4.146)	-
Esophagus	1	0.458 (0.281, 0.746)	1.000	0.458 (0.281, 0.746)	-
Head and neck	8	0.592 (0.495, 0.710)	<0.001	0.661 (0.430, 1.017)	0.366
Larynx	1	0.089 (0.025, 0.323)	1.000	0.089 (0.025, 0.323)	-
Lung	3	0.628 (0.497, 0.794)	0.519	0.628 (0.497, 0.794)	0.725
Stomach	2	0.181 (0.090, 0.364)	0.239	0.176 (0.077, 0.405)	-
Criteria					
<50%	3	0.475 (0.348, 0.648)	0.009	0.466 (0.222, 0.981)	0.959
50%	23	0.560 (0.509, 0.617)	<0.001	0.545 (0.457, 0.650)	0.320
>50%	2	0.618 (0.430, 0.889)	0.801	0.618 (0.430, 0.889)	-

CI, confidence interval.

## Data Availability

Not applicable.

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
