# Peer review of "Significance of Tumor–Stroma Ratio (TSR) in Predicting Outcomes of Malignant Tumors"

_medicina, 2023, doi:10.3390/medicina59071258_

Round 1
Reviewer 1 Report
Pyo et al. are conducting a meta-analysis on the impact of Tumor-stroma ratio (TSR) on Overall Survival (OS) and Disease-free Survival (DFS). Various types of tumors are comprehensively examined, and the significance of TSR is discussed in the manuscript. While I believe this paper adequately addresses the importance of TSR, there are some major points to be revised.
Major points
2. Materials and methods
2.1. Published Study Search and Selection Criteria
・The term "or" included in the search keywords and it does not represent an actual keyword input but rather indicates an "Either/or” search was performed. Please revise the main text to easily understand these differences.
・The quality of studies such as the Newcastle-Ottawa Scale has not been evaluated in the manuscript. Please add an evaluation and present it in a tabular format or any other suitable form. (Peng et al. Eur Neurol. 2023 Mar 27. doi: 10.1159/000530208.)
3. Results
3.1. Selection and Characteristics of the Studies
・There are several studies included in the literature that have different criteria for the Threshold of TSR. While it would be appropriate to include these studies if the number of eligible literatures are limited. However, the number of collected studies is considered sufficient for the meta-analysis in this case. Therefore, I believe it is appropriate to exclude the studies with different Threshold values (such as Threshold 40%) and proceed with the analysis.
3.2. Prevalence of high Tumor-Stoma Ratio
・The manuscript discusses heterogeneity. High heterogeneities were observed in several cancer types (Table 2). For cases with low heterogeneity, a fixed-effect model is considered appropriate (Sun et al. Clin Transl Oncol. 2023 Jun;25(6):1830-1843. doi: 10.1007/s12094-023-03080-1.).
Author Response
Pyo et al. are conducting a meta-analysis on the impact of Tumor-stroma ratio (TSR) on Overall Survival (OS) and Disease-free Survival (DFS). Various types of tumors are comprehensively examined, and the significance of TSR is discussed in the manuscript. While I believe this paper adequately addresses the importance of TSR, there are some major points to be revised.
Major points
- Materials and methods
2.1. Published Study Search and Selection Criteria
・The term "or" included in the search keywords and it does not represent an actual keyword input but rather indicates an "Either/or” search was performed. Please revise the main text to easily understand these differences.
Response) As pointed out by the reviewer, we revised the manuscript using actual keywords.
・The quality of studies such as the Newcastle-Ottawa Scale has not been evaluated in the manuscript. Please add an evaluation and present it in a tabular format or any other suitable form. (Peng et al. Eur Neurol. 2023 Mar 27. doi: 10.1159/000530208.)
Response) As pointed out by the reviewer, we evaluated eligible studies using the Newcastle-Ottawa Scale, and the results are presented in supplementary Table S2.
- Results
3.1. Selection and Characteristics of the Studies
・There are several studies included in the literature that have different criteria for the Threshold of TSR. While it would be appropriate to include these studies if the number of eligible literatures are limited. However, the number of collected studies is considered sufficient for the meta-analysis in this case. Therefore, I believe it is appropriate to exclude the studies with different Threshold values (such as Threshold 40%) and proceed with the analysis.
Response) Meta-analysis is a method of combining results from different methods to obtain a unified result, and analyzing the heterogeneity through a subgroup analysis. The criteria for high or low TSR varies across individual articles, and we performed a meta-analysis of the difference between high and low TSR. The reviewer's comments are discussed further as they may be a limitation of this meta-analysis as below: .
Fourth, we were unable to conduct analyses by criteria subgroup for high TSR for each cancer type due to insufficient information.
3.2. Prevalence of high Tumor-Stoma Ratio
・The manuscript discusses heterogeneity. High heterogeneities were observed in several cancer types (Table 2). For cases with low heterogeneity, a fixed-effect model is considered appropriate (Sun et al. Clin Transl Oncol. 2023 Jun;25(6):1830-1843. doi: 10.1007/s12094-023-03080-1.).
Response) Basically, each study has a different population, so this heterogeneity is inherent. Therefore, a random-effects model was used to analyze the data.
Reference
Michael Borenstein, Larry V. Hedges, Julian P.T. Higgins, Hannah Rothstein. Introduction to Meta-Analysis, 2nd Edition. Publisher: Wiley.
Reviewer 2 Report
Present article by Pyo et al titled "Significance of Tumor-stroma Ratio in Predicting clinical outcome of malignant tumors" is well written. However, few comments that I have are:
The introduction section is very less informative.
Table 1. What does "Criteria" column signify here.
The data is only represented in tabular form. The graphical representation of some important findings will make a good impact.
The conclusion is found to be contradictory as High TSR has both protective and negative effects specifically on survival. Therefore, it is important to mention in conclusion which malignancies falls the protective role of TSR, and which shows opposite trend.
English language is fine only minor editing is required.
Author Response
Present article by Pyo et al titled "Significance of Tumor-stroma Ratio in Predicting clinical outcome of malignant tumors" is well written. However, few comments that I have are:
The introduction section is very less informative.
Response) As the reviewer pointed out, we added the description in the introduction section. As our study evaluated a range of cancers, we minimized the detailed description of existing results for different cancers. Instead, we added the description based on our previous TSR research. In addition, the need for this meta-analysis study is further explained.
Table 1. What does "Criteria" column signify here.
Response) This is the criteria for dividing TSR high and low. We changed from "criteria" to "criteria for high TSR" to avoid misunderstandings.
The data is only represented in tabular form. The graphical representation of some important findings will make a good impact.
Response) As the reviewer points out, our results are presented in a table. The main results can be showed by switching to graphics, i.e. forest plot. However, it had to create one forest plot for each row in the table, which made it difficult to compare the data, so we provided all the results in a table.
The conclusion is found to be contradictory as High TSR has both protective and negative effects specifically on survival. Therefore, it is important to mention in conclusion which malignancies falls the protective role of TSR, and which shows opposite trend.
Response) Our results show that endometrial and pancreatic cancers are correlated with a poor prognosis. We added this description in the revised manuscript.
Round 2
Reviewer 1 Report
Summary
This paper shows significant improvement as a result of revision. However, there are still some areas that require further changes.
Major points
Materials and methods
2.1. Published Study Search and Selection Criteria
・The quality of studies such as the Newcastle-Ottawa Scale has not been evaluated in the manuscript. Please add an evaluation and present it in a tabular format or any other suitable form. (Peng et al. Eur Neurol. 2023 Mar 27. doi: 10.1159/000530208.)
The point regarding the quality assessment of studies using the Newcastle-Ottawa Scale has been addressed by the authors as a Supplementary table. However, the Newcastle-Ottawa Scale is a critical factor for evaluating the quality of research and should not be omitted from the main text when conducting a meta-analysis. Therefore, please present it as a formal table in the main body of the manuscript instead of in the Supplementary materials.
Furthermore, the authors have assigned only 0 or 1 for the "Comparability" domain of the Newcastle-Ottawa Scale. The "Comparability" domain should be scored on a scale of 0-2, considering the adjustment for confounders and other important factors. Does this mean that the most important confounding factors and other elements have not been adequately adjusted for?
Based on the authors' statement that this study has significant heterogeneity, if the scores on the Newcastle-Ottawa Scale are low, I believe this paper cannot be accepted.
Author Response
This paper shows significant improvement as a result of revision. However, there are still some areas that require further changes.
Major points
Materials and methods
2.1. Published Study Search and Selection Criteria
・The quality of studies such as the Newcastle-Ottawa Scale has not been evaluated in the manuscript. Please add an evaluation and present it in a tabular format or any other suitable form. (Peng et al. Eur Neurol. 2023 Mar 27. doi: 10.1159/000530208.)
Response) As pointed out by the reviewer, we evaluated based on Peng’s report.
The point regarding the quality assessment of studies using the Newcastle-Ottawa Scale has been addressed by the authors as a Supplementary table. However, the Newcastle-Ottawa Scale is a critical factor for evaluating the quality of research and should not be omitted from the main text when conducting a meta-analysis. Therefore, please present it as a formal table in the main body of the manuscript instead of in the Supplementary materials.
Response) As pointed out by the reviewer, we changed from supplementary table 2 to table 2 and added the table in the manuscript.
Furthermore, the authors have assigned only 0 or 1 for the "Comparability" domain of the Newcastle-Ottawa Scale. The "Comparability" domain should be scored on a scale of 0-2, considering the adjustment for confounders and other important factors. Does this mean that the most important confounding factors and other elements have not been adequately adjusted for?
Response) As pointed out by the reviewer, we evaluated the “Comparability” domain and changed the Table.
Based on the authors' statement that this study has significant heterogeneity, if the scores on the Newcastle-Ottawa Scale are low, I believe this paper cannot be accepted.
Response) Based on the evaluation by the Newcastle-Ottawa Scale, there was no excluded articles.
Round 3
Reviewer 1 Report
The manuscirpt revised well, and I think the current revision can be accepted for publication.